# Community participation enhanced prenatal mental health through strengthening peers' support and partners' communication in Chinese mothers: A cross-sectional study

**Shanshan An**, **Sheng Sun**¤*

Department of Medical Sociology, Jiangnan University, Wuxi, China

¤ Current Address: Department of Medical Sociology, Jiangnan University, Wuxi, Jiangsu, China
* 45060274@qq.com

## Abstract

This study investigated the relationship between extensive community participation and prenatal mental health, focusing specifically on the mediating roles of peer support and partner communication. A cross-sectional survey was conducted in Jiangsu, China, involving 1,705 pregnant participants. Structural Equation Modeling was employed to examine how offline peer support, online peer support, and partner communication mediate the association between community participation and prenatal mental health. Among the 1,705 participants (Mean age = 29.57, SD age = 3.70, Max age = 43, Min age = 16), 1,000 (58.7%) were giving birth for the first time, and 975 (57.3%) were local residents. The mean score for community participation was 4.43, while that for prenatal mental health was 4.21. These findings indicate that extensive community participation does not directly affect prenatal mental health. Instead, peer support and partner communication serve as parallel mediators, while social media and partner communication function as sequential (chain) mediators. These results underscore the critical need to enhance both the quantity and quality of partner-centered maternal support networks by leveraging peer groups and digital platforms. Strengthening these relational pathways can facilitate more effective communication between partners and ultimately contribute to improved maternal mental health during pregnancy.

## Introduction

Preventing prenatal mental health (PMH) problems is essential because of their negative impacts on women and their families. Prenatal mental health is typically categorized into two types: experiential and pathological imbalance. Experiential imbalance, which is more prevalent than pathological imbalance in developing countries such as China, often precedes and may develop into pathological conditions [1,2]. Some

**Data availability statement:** We recognize the significance of data sharing in promoting research transparency and replicability. However, due to the sensitivity of the data and our ethical commitment to the research participants, we have opted to impose certain limitations. The raw data we analyzed contains potentially identifiable and sensitive maternal information, even when encoded. In accordance with the ethics agreement approved by the ethics committee, we ensure the anonymity of participants and the confidentiality of the data. Access to the data is available through the Medical Ethics Committee of Jiangnan University (JNU20211217IRB01) (zhouyizhou@jiangnan.edu.cn) for researchers who meet the criteria for accessing confidential data. We have included a detailed introduction to the data collection and analysis processes in the research methods section to equip other researchers with the necessary information. (see lines 545-558) We believe this information equips other researchers with the methodological tools necessary to evaluate, compare, and replicate our findings without compromising the rights and privacy of the participants.We hope this revision clarifies the ethical and procedural limitations on data sharing, while also demonstrating our commitment to research transparency.

**Funding:** This research was supported by the National Social Science Fund of China (21CSH078). The contents of this manuscript are solely the responsibility of authors and don't necessarily represent the official view of NSSFC.The funders had no role in study design, data collection and analysis, decision to publish, or preparation of the manuscript.

**Competing interests:** No potential conflict of interest was reported by the author(s).

researchers argue that experiential imbalance is difficult to diagnose due to several factors in developing countries, including mistranslations of key psychological terms, variations in diagnostic thresholds across measurement tools [3], differing levels of mental health awareness across countries and cultures [4], and structural and cultural differences among various population groups [5]. From a social health perspective, the mental well-being of pregnant women is influenced not only by individual biological factors but also by external factors, such as cultural background, social support networks, and healthcare systems. Experiential imbalance, as an external manifestation of mental health issues, may evolve into clinical mental disorders; however, it is often initially expressed through emotional experiences, such as anxiety and loneliness [6]. Given these considerations, this study focuses on prenatal mental health as an experiential imbalance. Previous research has demonstrated that social support and community participation play important roles in mitigating psychological imbalances among pregnant women [7,8]. Social support can enhance positive social interactions, reduce negative emotions, and lower the risk of adverse pregnancy and birth outcomes [9]. Similarly, community participation can help pregnant women identify emerging psychopathological symptoms and provide them with a safe space to express emotional distress [10]. However, some studies caution that community involvement may inadvertently lead to negative consequences such as fostering feelings of competitive pressure or social isolation among pregnant participants [11].

Social support is a vital resource for individuals, offering external assistance that can help reduce social pressure, improve mental well-being, and promote personal growth [12]. This support is broadly defined as social resources (e.g., emotional, informational, and instrumental support) that individuals perceive as available when needed or that have actually been provided to them by their social systems, encompassing both formal and informal support [13–15]. Furthermore, it can be divided into perceived and actual support, depending on an individual's subjective perception [14], and can then be analyzed based on social structure, including community ties, social networks, and intimate ties. These components represent different layers of social relationships: outer support (participation in formal organizations), intermediary support (social networks), and inner support (close relationships), each potentially influencing mental health outcomes [16,17].

Cultural factors, including cultural adaptation, social discrimination, and stigma surrounding mental illness may partially influence the development of prenatal psychological issues [18]. Interpersonal communication in Chinese society is characterized by "Chaxu Geju", also known as the differential mode of association [19,20]. This concept aims to describe the pattern of close and distant relationships among individuals [20]. Each person is situated at the center of interconnected circles representing their relationships, with close bonds in the innermost circle and broader community involvement in the outer circle [19,20]. Building on social relations in China, this study classified support into inner support, intermediary support, and outer support. Outer support, which involves contact with organizations, such as hospitals, communities, and governments, plays a crucial role in protecting mental health. Intermediary support, such as unpaid help from social networks, peers, or online platforms,

also contributes significantly. Inner support, on the other hand, comes from close relationships with partners and friends. However, there is a lack of research on how to effectively coordinate support to help mothers achieve balance among the different types of support.

**Could three types of support improve prenatal mental health: based on current evidence**

Research has shown that three types of support can assist mothers in managing prenatal anxiety and enhancing their health literacy [7,8]. Furthermore, social support is thought to enhance positive social interactions, mitigate negative emotions, and ultimately reduce the likelihood of adverse pregnancy and birth outcomes [9]. Additionally, it can equip pregnant women with effective coping strategies to navigate stressful situations [21].

Regarding the inner support dimension, women who receive enhanced support from family and friends exhibit lower levels of anxiety [22,23]. In China, the longstanding tradition of family support encompasses practices such as postpartum confinement, which serves as a significant form of assistance. However, recent research has revealed potential conflicts within family support networks, including intergenerational tensions and disputes between mothers-in-law and daughters-in-law, which can inadvertently increase psychological stress in mothers [24]. Letourneau et al. (2007) also noted that although support from close friends and family is crucial, it may not always be sufficient, as some women reported feeling overwhelmed and made adjustments to cope with their stress [25].

The relationship between a woman and her partner is frequently regarded as a more significant protective factor in close relationships than that with parents and friends [26]. Partner communication plays a crucial role in intimate relationships and is typically included in healthy relationship strategies to prevent mental health issues. Effective communication can reduce misunderstandings and loneliness, build mutual trust, and help alleviate negative emotions like anxiety and depression in pregnant women [27]. Research indicates that increased emotional support from partners can encourage pregnant women to actively seek social support, leading to better utilization of available resources and improved health outcomes [27]. Positive feedback from a partner regarding one's abilities can boost maternal confidence, influence self-efficacy beliefs, and ultimately contribute to maternal psychological well-being and improved health outcomes in high-risk women [28].

Regarding the intermediary support dimension, the rise of social media, changes in traditional family structures, and shrinking health and social care services have expanded the concept of trust to encompass broader social relationships beyond immediate family members [29]. Mothers often seek solutions to their physical and mental health issues through online platforms and pregnant women's groups, perceiving them as more accessible and reliable than traditional sources of support such as family members [30]. Mothers' groups can play a vital role in helping mothers recognize the symptoms of psychopathology [10]. Mothers experiencing prenatal depression often share emotional struggles with these groups [10]. Online social networks also provide mothers with information and support, boosting their confidence in childbirth and parenting but may lead to feelings of isolation and competition among some mothers [11].

As defined by Lin (2008), outer support encompasses an individual's level of involvement in community activities [17]. Formal social participation holds significant importance in the realm of health promotion, serving as a potential gateway to various social constructs that can enhance essential protective factors for health [31]. Engaging in broader social participation has been linked to increased social support, which in turn can effectively alleviate anxiety symptoms [32]. Furthermore, such participation can foster social integration, nurture a sense of belonging, and foster positive interpersonal relationships [33]. This interconnectedness also facilitates the exchange, persuasion, and support of health information, directly influencing individuals to adopt healthier behaviors [34].

**Social support network and partner communication: potential mechanisms of change for the relationship between community participation and PMH**

The findings suggest that all forms of support have a direct impact on the mental health of pregnant and postpartum women. Community participation has emerged as a fundamental protective factor for prenatal mental health in contemporary

societies. In the context of Chinese society, government agencies play an active role in establishing formal organizations that facilitate connections between pregnant and postpartum women and the community. These include channels for psychological counseling and crisis intervention, medical security systems and institutions, and community health and education platforms. Research indicates that Chinese mothers' voluntary engagement in interactive behavior with medical institutions does not directly enhance their mental health outcomes [35]. The broader the social participation, the more positive the impact, which warrants further investigation. The general proposal is that each inner layer is contingent upon the outer layers, with each outer layer of linkages affording the opportunity to construct inner layer linkages [16]. However, Lin proposed that binding is a significant source of social resources in East Asian societies, where social relations have expanded from family relations [17]. This implies that the level of community engagement among mothers could affect their postpartum mental health through intermediary factors. The primary mediating factor was partners' communication, while social network components were seen as secondary influences. Based on these findings, this study proposes the following hypothesis:

H1: Communication participation does not directly impact PMH.

H2: Peer support, whether facilitated through offline groups or social media platforms, serves as a mediating factor in the relationship between community participation and PMH.

H3: A serial multiple mediation effect exists, where offline peer groups provide support and partner communication influences the relationship between community participation and PMH.

H4: A serial multiple mediation effect is observed when peer groups on social media provide support, and partner communication affects the relationship between community participation and PMH.

## Methods

### Study design

We conducted a cross-sectional survey on a representative sample of pregnant women in China. From February 2022 to October 2022, participants were recruited through quota sampling in three obstetric hospitals in Jiangsu Province. The three hospitals had different characteristics: A was a specialized hospital for maternal and child healthcare, B was a critical care center for pregnant and postpartum women, and C was the obstetric center of a university-affiliated hospital. The inclusion criteria were as follows: (1) mothers who were more than 12 weeks pregnant; (2) Residing in Wuxi City, Jiangsu Province and having established maternal and child health records in the obstetrics department; and (3) Fluent in Chinese. The research team initially conducted a pre-survey involving 30 pregnant women, ensuring that the participants were not included in the subsequent formal survey. The pre-investigation process consisted of six steps: selecting the instrument, expert review for validity, applying for ethics committee approval, selecting respondents, executing the pre-test, and reviewing results. After confirming the reliability and validity of all instruments as well as the comprehensibility of all questions, the research team proceeded to the formal investigation. Eligible participants were guided by the research team to view the online research information form and consent form while waiting for prenatal examinations, and were provided with a survey file package (survey questionnaire, pen). Women were required to complete the survey questionnaire and return it to the prenatal care nursing station. The survey questionnaire, titled Maternal Depression Generation Trajectory and Social Support Research,'included demographic data, the PMH scale, perceived social support, actual social support, partners' communication, social stigma, community participation, social support satisfaction, care-seeking behavior, social trust, and other items. The participants took approximately 20 to 30 minutes to complete the survey. This study specifically focused on four variables: mothers' PMH, social support from peers (offline peer groups or online platforms), partners' communication, and community participation. Other elements were included in the separate studies. This study was conducted in accordance with the guidelines of the Declaration of Helsinki and approved by the Medical Ethics Committee of the University of Jiangnan, Wuxi (JNU20211217IRB01, date of approval December 17, 2021). All the participants signed an informed consent form at the beginning of the survey. They were guaranteed anonymity and were allowed to discontinue the survey at any time.

## Participants

The credibility of survey research findings is significantly influenced by the response rate [36]. Previous studies have indicated that errors can arise from non-responses as well as from respondents providing inaccurate answers [37]. In this study, effective recycling of data necessitates two conditions: (1) the questionnaire must be fully completed without any missing responses and (2) respondents must refrain from intentionally providing incorrect answers. To ensure sufficient statistical power, we conducted a power analysis using G*Power ($f^2 = 0.15$, $\alpha = 0.05$, power $= 0.95$, two-tailed), indicating a minimum sample size of 107. Our final sample far exceeded this threshold. A quota sampling strategy was implemented in collaboration with three partner hospitals to ensure demographic diversity and model robustness. Annual births at Hospitals A, B, and C are approximately 10,000, 5,300, and 5,100 respectively. We adopted a cross-quota method stratified by hospital and gestational stage (50/50 early vs. late pregnancy), with a 10/100 sampling ratio. This yielded 2,030 sampled participants: 1,000 from Hospital A, 530 from Hospital B, and 500 from Hospital C. Ultimately, we collected 948 cases from Hospital A, 487 cases from Hospital B, and 464 cases from Hospital C, resulting in a total of 1,899 distributed questionnaires in this survey, of which 1899 were recovered and 1705 were validated, resulting in a valid response rate of 89.8%. Among the 1705 participants, the mean age was 29.57 years (SD = 3.71, range 16–43). The average gestational cycle was 30.58 weeks, with durations ranging from 12 weeks to 41 weeks. Of the participants, 1,000 (58.7%) gave birth for the first time, 487 (28.6%) for the second time, and 218 (12.7%) for the third time (See Table 1 for more information). The majority of the participants had a bachelor's degree or higher (81.5%), and over half of the families had a monthly income of less than 15,000 yuan (58.9%).

## Measurement instrument

**Community participation.** This variable follows Lin's research methodology to assess community connections through involvement in community organizations [16]. Participants were requested to disclose the number of organizations with which they were affiliated or received assistance during pregnancy. These organizations span various sectors, such as work units, party organizations, trade unions, hospitals, community self-organizations, and other groups. The findings indicate that expectant mothers were members of at least 0 organizations and up to 11 organizations, with an average of 4.43.

**Table 1. Description of participant demographics.**

| Characteristics | Categories | N(%)/ Mean(S.E.) |
|---|---|---|
| Age (years) | | 29.57(3.71) |
| Gestational age (weeks) | | 30.58(8.09) |
| Local residents | Yes | 975(57.3) |
| | No | 730(42.7) |
| Parity | 1st | 1000(58.8) |
| | 2nd | 487(28.6) |
| | 3rd or more | 213(12.5) |
| Education level | High school or below | 315(18.5) |
| | College/university | 1231(72.2) |
| | postgraduate or above | 157(9.3) |
| Household income (monthly) | Less than10000 | 514(30.3) |
| | 10000-15000 | 482(28.4) |
| | 15001-20000 | 358(21.1) |
| | More than 20000 | 343(20.2) |
| Total | | 1705(100) |

## Mother's PMH

The revised Body Mind-Spirit Well-Being Inventory (BMSWBI) was utilized to assess the mental health of Chinese mothers [38]. Given the somatization of mental health issues in the Chinese population, it is crucial to recognize that mental health is a multifaceted construct encompassing various elements, such as thoughts, feelings, values, ethics, beliefs, and their interconnectedness with the physical body. To obtain comprehensive insight into the mental health status of Chinese mothers, the author chose to utilize a dynamic and systemic scale tailored for this purpose [39]. The final scale comprised 20 items rated on a 6-point scale (1 = completely inconsistent, 6 = completely consistent). These items were found to cluster into three subfactors (KMO = 0.90, $p < 0.000$) representing the body ($\alpha = 0.82$), mind ($\alpha = 0.81$), and spirit ($\alpha = 0.78$) with strong reliability ($\alpha = 0.87$).

## Social support from peers

This study measured the support received from offline peer groups, specifically pregnant women, and social media [40]. The dimensions of support were categorized into emotional, instrumental, and informational support. Instrumental support refers to tangible assistance that individuals receive, such as financial help or aid with daily tasks [41]. For example, one might say, "When I have a need, I can receive assistance with daily chores." Informational support involves the provision of relevant knowledge, advice, or guidance aimed at helping individuals cope with challenges or make informed decisions [41]. An example of this is, "I am provided with comprehensive and truthful information regarding pregnancy, childbirth, and parenting." Emotional support encompasses expressions of empathy, care, reassurance, and trust, offering individuals opportunities for emotional expression and psychological relief [41]. For instance, one might express, "I can confide in others and discuss the joys and sorrows of pregnancy." Participants evaluated the level of emotional, instrumental, and informational support received from each source using a 6-point Likert scale, where 1 indicates 'not at all' and 6 indicates 'always.'

## Partners' communication

The Initiator Style Questionnaire (ISQ) was developed by Denton and Burleso [42]. The questionnaire prompts mothers to assess their inclination to initiate or avoid discussions about relationship issues with their partners. Sample questions included statements such as I frequently communicate my feelings to my partner about our relationship dynamics, I enjoy addressing conflicts that arise in our relationship with my partner, and 'My partner persists in discussing relationship issues with me until they are resolved, among others. The final analysis employed the average score to indicate the communication orientation of partners, ranging from 1 to 6. A higher score indicates a stronger communication orientation. The scale demonstrated good reliability with a Cronbach's alpha coefficient of 0.82.

## Demographic variables

Variables included age, education, household registration, household income, and pregnancy duration.

## Data analyses

This study was conducted in two phases. The first phase involved descriptive, exploratory factor, and reliability analyses of demographic and sociological characteristics. In the second phase, the predictive power of community participation in PMH was assessed using Structural Equation Modeling (SEM). SEM was chosen over correlation analysis or multiple regression analysis due to its ability to test an overall model rather than individual model coefficients and incorporate multiple relevant and mediating variables [43]. The analysis focused on two main aspects: examining the parallel mediating effects of peer and partner communication on the relationship between community participation and PMH and verifying the mediating effect of peer groups or networks on the relationship between community participation and partners'

communication while considering the impact of peers' support factors on partners' communication. Multiple serial mediations were assessed using bootstrapping. Statistical analysis was conducted using SPSS 25.0 and AMOS 25.0.

## Results

### Correlations of the key variables

Table 2 displays the mean, standard deviation, and correlations of the key variables. The dependent variable (DV) consisted of three components: physical, mental, and spiritual health. The independent variable (IV) was community participation. The mediating variable of offline peer support comprises three aspects: peer groups' emotional support, informational support, and instrumental support. Similarly, the mediating variables of online peer support include social media's emotional, informational, and instrumental support. Additionally, the mediating variable of partner communication encompassed expression tendency, avoidance tendency, and listening and communication skills. This analysis focuses on the correlations between the dependent and independent variables as well as the three mediating variables.

The total score for mother's PMH (DV) was 4.23 points. Among the different aspects, physical health had the lowest mean score (M = 4.07, SD = 1.11), spiritual health scored at an intermediate level (M = 4.12, SD = 0.82), and emotional health had the highest score (M = 4.51, SD = 0.99). Community participation (IV) received a rating of 4.43. Practical support from the peer groups received an overall score of 3.04. Instrumental support scored the lowest (M = 2.46, SD = 1.44), emotional support in the middle (M = 3.25, SD = 1.48), and informational support in the highest (M = 3.42, SD = 1.47). Regarding social media support, the overall score was 3.36. Instrumental support had the lowest score (M = 2.58, SD = 1.57), followed by emotional support (M = 3.62, SD = 1.54), and informational support (M = 3.87, SD = 1.49). Finally, in terms of partners' communication, the overall score was 4.39. Avoidance tendency was the lowest (M = 4.13, SD = 1.03), listening and communication were in the middle (M = 4.29, SD = 1.05), and expression tendency had the highest scores (M = 4.73, SD = 0.89). Local mothers reported significantly higher levels of PMH (F = 13.56), enhanced spiritual well-being (F = 13.77), and greater social (F = 35.09), emotional (F = 11.55), informational (F = 19.79), and instrumental support (F = 31.25, $p < .001$). No differences were observed in body well-being or community participation. Across hospitals, body well-being (F = 3.89, $p < .10$) and instrumental support (F = 3.10, $p < .10$) showed modest differences, with hospital C scoring higher. Community participation exhibited the largest difference (F = 9.70, $p < .001$), with participants from C being more involved than those from hospitals A and B. First-time mothers reported significantly greater community participation (F = 13.56), emotional support (F = 12.92), and instrumental support (F = 26.81, $p < .001$) than multiparous mothers. Post-hoc tests revealed moderate effects, particularly for instrumental support (e.g., d = 0.538 compared to third-time mothers).

Based on the correlations among the key variables, several meaningful relationships emerged among the PMH variable, social support, partners' communication, and community participation. As anticipated, mental and physical health exhibited a moderate correlation (r = 0.585, p < .001), indicating a robust link between psychological and physiological well-being. Spiritual health demonstrated moderate positive correlations with both mental health (r = 0.306, p < .001) and community participation (r = 0.176, p < .001), suggesting that spiritual well-being may be enhanced through mental resilience and communal engagement. All three types of online peer support (emotional, informational, and instrumental) were significantly associated with spiritual health, with informational support exhibiting the strongest relationship (r = 0.262, p < .001). However, online peer support generally displayed weaker associations with physical and mental health than peer group support, reflecting potential differences in perceived reliability or intimacy. The tendency to express emotions was positively correlated with all health domains, particularly mental health (r = 0.249, p < .001) and spiritual health (r = 0.430, p < .001).

A serial mediation model was used to examine the mediating roles of peer support and partners' communication in the relationship between community participation and PMH. The findings in Table 3 demonstrate that both measurement models fit the data well. Model A, which considers peer groups as the peer support factor, displayed

**Table 2. Means, standard deviations, and correlations of the key variables.**

| | M | SD | Physical health | Mental health | Spiritual health | Community participation | Peers groups emotional | Peers groups informational | Peers groups instrumental | Social media emotional | Social media informational | Social media instrumental | Expression tendency | Avoidance tendency |
|---|---|---|---|---|---|---|---|---|---|---|---|---|---|---|
| Physical health | 4.07 | 1.11 | 1 | | | | | | | | | | | |
| Mental health | 4.51 | 0.99 | 0.585 *** | 1 | | | | | | | | | | |
| Spiritual health | 4.12 | 0.82 | 0.234 *** | 0.306 *** | 1 | | | | | | | | | |
| Community participation | 4.43 | 2.65 | 0.095 * | 0.145 *** | 0.176 *** | 1 | | | | | | | | |
| Peers groups_emotional | 3.25 | 1.48 | 0.029 | 0.046 | 0.241 *** | 0.126 *** | 1 | | | | | | | |
| Peers groups_informational | 3.42 | 1.47 | 0.065 ** | 0.079 ** | 0.262 *** | 0.141 *** | 0.739 *** | 1 | | | | | | |
| Peers groups_instrumental | 2.46 | 1.44 | 0.074 ** | 0.059 * | 0.185 *** | 0.137 *** | 0.538 *** | 0.556 *** | 1 | | | | | |
| Social media_emotional | 3.62 | 1.54 | −0.008 | 0.01 | 0.093 ** | 0.08 *** | 0.419 *** | 0.36 *** | 0.304 *** | 1 | | | | |
| Social media_informational | 3.87 | 1.49 | −0.002 | 0.019 | 0.111 *** | 0.088 *** | 0.32 *** | 0.417 *** | 0.266 *** | 0.626 *** | 1 | | | |
| Social media_instrumental | 2.58 | 1.57 | 0.012 | 0.027 | 0.061 * | 0.097 | 0.31 *** | 0.328 *** | 0.615 *** | 0.48 *** | 0.458 *** | 1 | | |
| Expression tendency | 4.73 | 0.89 | 0.146 *** | .249** *** | 0.430 *** | 0.189 *** | 0.179 *** | 0.215 *** | 0.127 *** | 0.123 *** | 0.197 *** | 0.11 *** | 1 | |
| Avoidance tendency | 4.13 | 1.03 | 0.136 *** | .175** *** | 0.344 *** | 0.152 *** | 0.11 *** | 0.109 *** | 0.136 *** | 0.046 | 0.082 ** | 0.101 | 0.513 *** | 1 |
| Listening & communication | 4.29 | 1.05 | 0.194 *** | .271** *** | 0.209 *** | 0.105 | 0.056 * | 0.067 ** | −0.002 | 0.054 * | 0.068 ** | −0.007 | 0.282 *** | 0.308 *** |

*$p < .05$, **$p < .01$, ***$p < .001$.

Mediation analysis with structural equation modeling.

**Table 3. Multiple mediation model A and model B.**

| Paths | Estimate | | S.E. | C.R. |
|---|---|---|---|---|
| | B | β | | |
| Model A (peers' group as a social network) | | | | |
| Community participation →Peers' group | 0.076 | 0.162 | 0.012 | 6.266*** |
| Community participation → Partner communication | 0.053 | 0.206 | 0.007 | 7.223*** |
| Community participation→PMH | 0.007 | 0.059 | 0.004 | 1.98 |
| Peers' group→Partner communication | 0.122 | 0.222 | 0.017 | 7.203*** |
| Peers' group→PMH | 0.049 | 0.182 | 0.010 | 4.985*** |
| Partner communication→PMH | 0.334 | 0.678 | 0.039 | 8.509*** |
| Model B (peers through social media) | | | | |
| Community participation→Social media | 0.053 | 0.114 | 0.013 | 4.189*** |
| Community participation → Partner communication | 0.047 | 0.221 | 0.007 | 7.759*** |
| Community participation→PMH | 0.011 | 0.085 | 0.004 | 2.691*** |
| Social media→Partner communication | 0.102 | 0.185 | 0.018 | 5.764*** |
| Social media→PMH | −0.005 | −0.017 | 0.010 | −0.493 |
| Partner communication→PMH | 0.381 | 0.748 | 0.043 | 8.858*** |

Model A: χ²=145.478, df=29, CFI=.974, TLI=.960, RMSEA=.049. *p<.05, **p<.01, ***p<.001.

Model B: χ²=137.997, df=29, CFI=.970, TLI=.954, RMSEA=.047. *p<.05, **p<.01, ***p<.001.

$\chi2=145.478$, df=29, CFI=0.97, TLI=0.96, RMSEA=0.05, while Model B, with social media as the peer support factor, yielded $\chi2=137.997$, df=29, CFI=0.97, TLI=0.95, RMSEA=0.05. The initial Model A results indicate that while increased community participation directly influences maternal PMH enhancement, this relationship is not statistically significant in certain groups: Model A (CR). =1.980, $p$=not significant), or Model B (C.R.=2.691, $p<0.05$), thus providing partial support for H1.

This study examined the mediating roles of peer support and partners' communication in the relationship between community participation and psychological well-being (PMH). In Model A, community participation significantly predicted two mediators, peer group support (β=0.162, p<0.001) and partners' communication (β=0.206, p<0.05). Furthermore, the direct effect of peer group support on partners' communication was significant (β=0.222, p<0.001). In Model B, incorporating social media for support and partner communication as sequential mediating variables resulted in the previously significant relationship between social media for support and PMH becoming non-significant (β=−0.017, p=not significant). This finding indicates that partners' communication fully mediates the association between social media for support and PMH, thereby providing support for H2.

Based on the results of the previous analysis, bootstrapping was used to further examine the mediating variables. The bootstrap method was employed to estimate the standard errors of the direct and indirect effects of the mediating variables and provide confidence interval values. The outcomes are detailed in Table 4 and Fig 1, illustrating the notable overall impact of community participation on the PMH (Std. estimate=0.032).

Notably, the indirect effects of community participation on PMH through partners' communication and peer groups on social media support were statistically significant (Std. estimate=0.025, p<0.001), with these mediating factors collectively explaining 78.1% of the total path. Furthermore, the mediating effect of peer groups contributed 45.0% of the association between community participation and partners' communication for support. Similarly, as shown in Table 4 and Fig 2, there was a significant overall influence of community participation on PMH (Std. estimate=0.035).

**Table 4. Test of the mediating effect by the bootstrap method.**

| Paths | Total effect | Direct effect | Indirect effect | CI |
|---|---|---|---|---|
| **Model A (peers' group offline as a social network)** | | | | |
| Community participation→ Peers' group | 0.076 | 0.076 | 0 | .108−.226 |
| Community participation→Partner communication | 0.062 | 0.053 | 0.009 | .158−.256 |
| Community participation→PMH | 0.032 | 0.007 | 0.025 | −.010−.118 |
| Peers' group→ Partner communication | 0.122 | 0.122 | 0 | .143−.281 |
| Peers' group→ PMH | 0.090 | 0.049 | 0.041 | .117−.243 |
| Partner communication→PMH | 0.334 | 0.334 | 0 | .587−.751 |
| **Model B (peers through social media as a social network)** | | | | |
| Community participation→ Social media | 0.053 | 0.053 | 0 | .056−.168 |
| Community participation→Partner communication | 0.062 | 0.057 | 0.005 | .162−.271 |
| Community participation→PMH | 0.035 | 0.011 | 0.024 | .013−.144 |
| Social media→ Partner communication | 0.102 | 0.102 | 0 | .129−.260 |
| Social media→ PMH | 0.034 | −0.005 | 0.039 | −.095−.039 |
| Partner communication→PMH | 0.381 | 0.381 | 0 | .669−.832 |

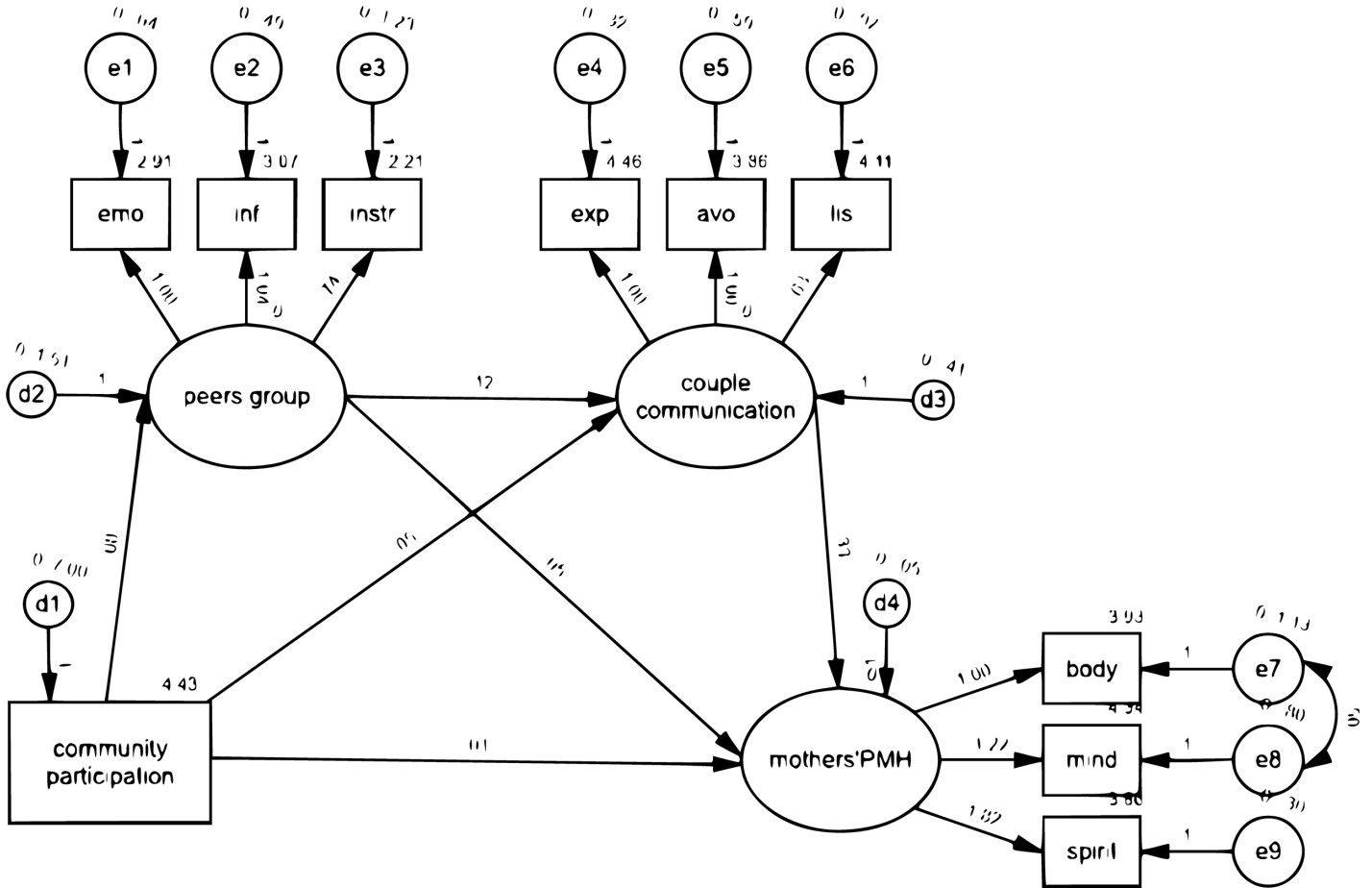

**Fig 1. Path analysis of community participation, PMH, peers' group, and partner communication.**

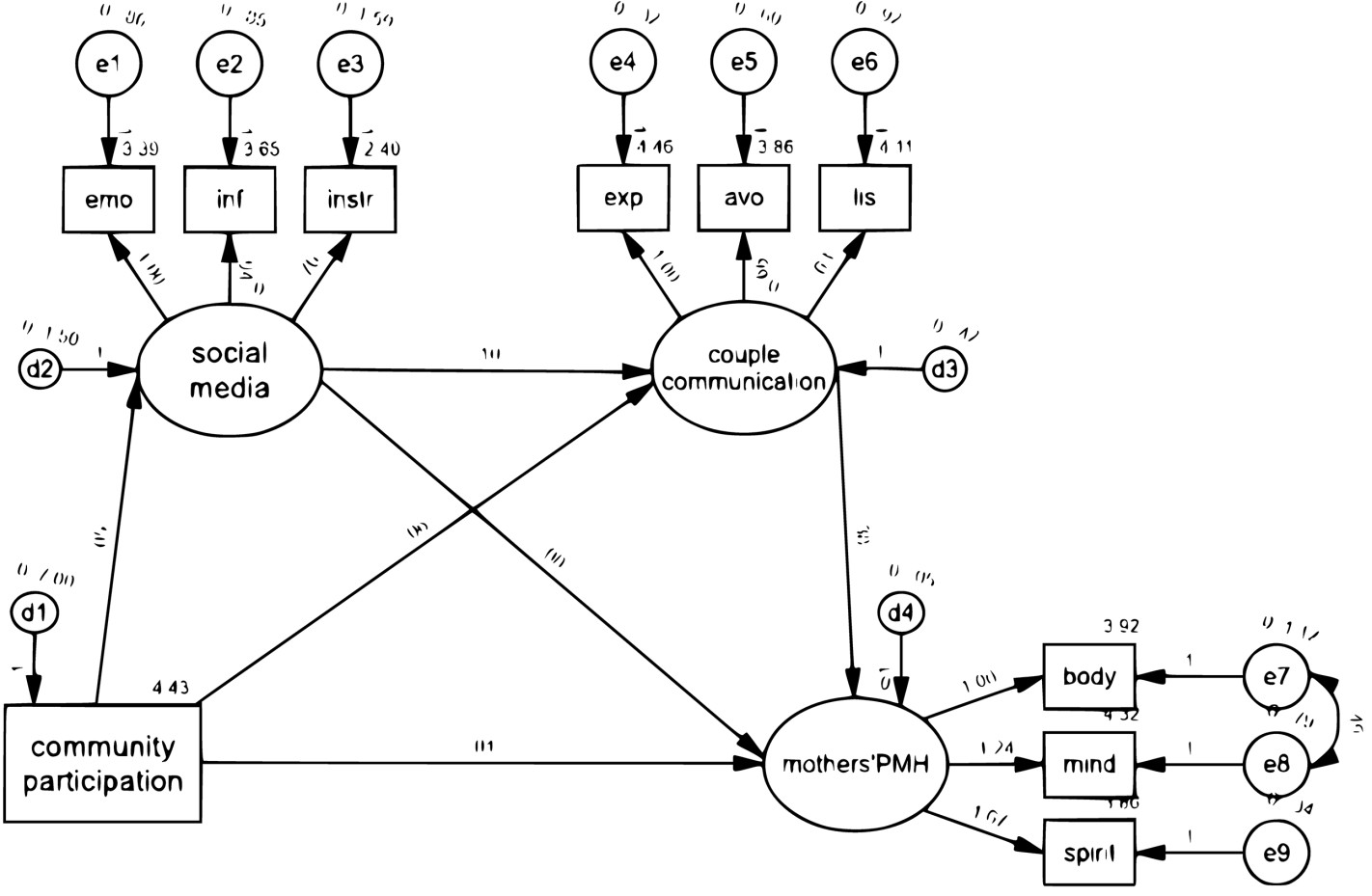

**Fig 2. Path analysis of community participation, PMH, social media, and partner communication.**

The indirect effect of community participation on PMH through social media support and partners' communication (Std. estimate = 0.024, p < 0.001; Std. estimate = 0.102, p < 0.001) was also found to be significant, with the mediating role of social media and partners' communication for support accounting for 71.4% of the path from community participation to PMH. Furthermore, the mediating effect of social media contributed 24.2% of the association between community participation and partners' communication for support. Therefore, H3 was supported, indicating that community participation impacts PMH through the parallel mediating effects of peer group support and partners' communication. Additionally, H4 was supported, suggesting that community participation influences PMH through the chain mediation effect of social media for support and partners' communication.

## Discussion

Few studies have examined the relationship between structural social support and PMH among Chinese mothers. Scholars suggest that cultural factors such as cultural adaptation, social discrimination, or mental illness stigma may partially influence this trend [18]. Drawing from Lin's (1999) social support theory and considering the ' ChaxuGeju' of Chinese society [16,20], we categorized structural social support into three levels: community participation at the outer layer, peer support at the intermediary level, and partners' communication at the inner level. These results suggest that community

participation may not directly impact the mental health of Chinese mothers. Despite varying degrees of participation in social support networks, the relationship between the number of support networks and mental health appears weak, which contrasts with the findings of previous studies [44]. This divergence could be attributed to systemic issues, such as public policy reforms and deficiencies in the medical system, which contribute to a lack of trust in external support among Chinese mothers [45]. Additionally, it is possible that stakeholders do not possess a comprehensive understanding of their interactions and relationships with one another [46].

The mediating effect test demonstrated that peer support and partner communication serve as mediators in the relationship between community participation and PMH. Our findings are consistent with those of previous studies that underscore the significance of fostering peer groups to enhance external support, intimate relationships, and individual well-being [47]. Increased community participation provides mothers with numerous opportunities to connect with their peers, allowing them to share and discuss their experiences of motherhood. Engaging in open conversations about pregnancy, childbirth, and postpartum experiences that deviate from societal norms not only alleviates feelings of isolation among women, but also facilitates a connection between mothers and healthcare professionals [48].

Social media support plays a crucial role in connecting external support and personal problems through partner communication. Chinese mothers often depend on online information for health-related decisions [30], but the prevalence of inaccurate maternal health information on the Internet can adversely affect their mental wellbeing [49]. Managing relationships on social media can also lead to social media fatigue [50]. Our research indicates that the positive effects of social media on maternal mental health are primarily realized through partner communication, highlighting the significance of such interactions in enhancing maternal well-being [51].In the context of 'mother and baby units,' fathers of unborn children are frequently marginalized during pregnancy and the postpartum period, often lacking the knowledge necessary for effective communication regarding their partners' health. Social media has the potential to engage fathers in the healthcare process, thereby facilitating communication and interaction between partners [52].

Support no longer relies solely on traditional family networks in the context of increasingly diversified community participation structures. Peers and the Internet have emerged as primary channels for Chinese mothers to seek assistance. This is particularly true for Chinese mothers with non-local household registration who lack effective family support, and thus require additional guidance from sources outside the family as a supplement. This trend is not unique to Chinese society but is prevalent in many developing countries. However, current policies, such as expert consensus on maternal mental health management, tend to overlook the significance of peers in promoting personal health. Instead, the focus is often on the utilization of outer and inner support by mothers. Thus, it is imperative to investigate how to optimize the expansion of peers as an effective support system that can lead to positive health outcomes. Partner communication plays a vital role in mediating the relationship between peers, particularly social media, and PMH. While it is common in Chinese society to use the internet or artificial intelligence for remote psychological support and mental health services for mothers, targeting individual mothers may not always result in positive outcomes. In fact, focusing solely on mothers can sometimes lead to a decline in mental health. Motherhood and childbirth involve identity transformation and recognition by both parents, and involving partners can help counteract potential negative effects. Strengthening partners' interactions through peers can enhance effective partner communication, ultimately improving maternal mental health. Therefore, it is essential to expand communication channels for partners and enhance the quantity and quality of partner-centered maternal support networks.

In considering the mediating variables, the effectiveness of 'narratives and sharing' in mediating the relationship between community participation and personal health was emphasized. Maternal self-doubt often emerges during pregnancy due to comparisons with past or ideal selves, other women, or unattainable gender ideals [53]. It is crucial for mothers to not only enhance their support networks but also articulate their experiences and emotions within supportive social circles. Shared experiences have a positive impact on healing and self-care, contributing to the development of self-worth and new identity [53]. It is imperative to move beyond individual factors and explore how adaptive interactions

among individuals with diverse life experiences, organizational structures, and technological environments can yield positive results.

At the intervention development level, it is essential to position the community as a central platform. Providing partners with various support options, integrating policy information and interventions, and leveraging peer groups and online resources to promote relationships among partners can significantly improve maternal mental health. We recommend that strengthening community mental health services can empower mothers to seek support earlier, utilize these services more effectively, become advocates within their communities, connect external and internal spheres, and foster meaningful relationships.

## Limitations

This study aims to examine the role of community involvement and social support in prenatal mental health, providing insights that may inform the design and implementation of supportive interventions. However, our study had some limitations.

First, the study relied solely on self-reported data from Chinese pregnant women, omitting perspectives from partners, family members, or healthcare providers. This limitation may affect the generalizability of the findings and overlook the important social dynamics. To address this issue, a follow-up qualitative study involving key members of the social network has been conducted and will be reported separately.

Secondly, although the Body–Mind–Spirit Well-Being Inventory was selected for its cultural relevance, it does not evaluate clinical symptoms, such as depression or anxiety. The lack of standardized diagnostic tools (e.g., the PHQ-9 and GAD-7) may hinder clinical interpretation. Future studies should consider integrating culturally grounded clinical measures.

Thirdly, the COVID-19 quarantine policy has significantly hindered our ability to collect cross-regional and large-scale data. Consequently, this study utilized regional samples and gathered data in clinical settings, which may limit our capacity to observe the evolution of mental health and support throughout pregnancy and the postpartum period. To address this limitation, the research team is currently conducting a longitudinal qualitative study that tracks 83 couples throughout pregnancy and postpartum to explore changes in mental health and support over time.

## Conclusion

This study offers novel insights into the nuanced interplay between structural social support and PMH in the Chinese context, underscoring the importance of moving beyond the surface-level metrics of support quantity. By highlighting the mediating roles of peer support and partner communication within a culturally embedded social framework such as "Chaxu Geju," our findings advocate for a paradigm shift from merely expanding community participation to cultivating emotionally resonant, trust-based, and dialogic relationships. This reconceptualization has significant implications for both policy development and intervention design. In an era in which traditional familial support structures are increasingly fragmented due to urbanization, migration, and shifting gender roles, the emergence of peer networks and digital communities presents both challenges and opportunities. Current maternal health frameworks in China—and in many other developing societies—remain predominantly mother-centric and medically oriented, often underestimating the transformative potential of mid-layer social actors such as peers and partners. Our findings suggest that these intermediary layers are not peripheral but foundational to the psychological well-being of mothers during pregnancy. Therefore, future interventions should strategically reposition community platforms to dynamic ecosystems for relational engagement. This includes institutionalizing peer-based programs within maternal healthcare services, incorporating narrative-sharing as a therapeutic and identity-building mechanism, and leveraging digital technologies not only to disseminate information but also to facilitate reciprocal communication and emotional intimacy—particularly between mothers and partners. Encouraging partner involvement through well-designed and culturally sensitive tools could significantly amplify the positive effects of social media and peer support, contributing to more holistic and sustainable maternal mental health outcomes.

# Acknowledgments

The authors are deeply grateful to all the pregnant women who generously participated in this study, and to the doctors and nurses of the hospital for their invaluable support and assistance throughout the investigation. We also sincerely acknowledge the reviewers for their insightful comments and constructive suggestions, which have significantly contributed to improving the quality of this manuscript.

# Author contributions

**Data curation:** Shanshan AN.

**Formal analysis:** Shanshan AN.

**Funding acquisition:** Shanshan AN.

**Investigation:** Shanshan AN.

**Methodology:** Shanshan AN, Sheng Sun.

**Writing – original draft:** Shanshan AN.

**Writing – review & editing:** Sheng Sun.

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
