## [Decision Letter · Decision Letter 0]

3 Jun 2025

Dear Dr. Sun,

Thank you for submitting your manuscript to PLOS ONE. After careful consideration, we feel that it has merit but does not fully meet PLOS ONE’s publication criteria as it currently stands. Therefore, we invite you to submit a revised version of the manuscript that addresses the points raised during the review process.

We look forward to receiving your revised manuscript.

Kind regards,

Fucai Lin, Ph.D.

Academic Editor

PLOS ONE

Journal Requirements:

4. In the online submission form, you indicated that The data that support the findings of this study are available on request from the corresponding author, Sun Sheng. The data are not publicly available due to restrictions e.g., their containing information that could compromise the privacy of research participants.

5. Please remove all personal information, ensure that the data shared are in accordance with participant consent, and re-upload a fully anonymized data set.

Reviewers' comments:

Reviewer's Responses to Questions

**Comments to the Author**

1. Is the manuscript technically sound, and do the data support the conclusions?

Reviewer #1: Yes

Reviewer #2: Yes

2. Has the statistical analysis been performed appropriately and rigorously?

Reviewer #1: Yes

Reviewer #2: Yes

3. Have the authors made all data underlying the findings in their manuscript fully available?

Reviewer #1: No

Reviewer #2: No

4. Is the manuscript presented in an intelligible fashion and written in standard English?

Reviewer #1: Yes

Reviewer #2: Yes

Reviewer #1: The study shows the importance of social support networks and effective communication between partners in reducing stress, anxiety, and isolation, which can contribute to better mental health outcomes among mothers. The research is significant as it may help promote long-term societal benefits, such as healthier families and empowered individuals, while informing policies and practices that integrate mental health into prenatal care. However, there are some issues that need to be addressed in the paper.

1. Clarify which 'father' you are referring to in this context: "Research indicates that increased emotional support from fathers…" .Is it the father of the pregnant woman or father of the unborn baby?

2. Some statements in the paper should be backed with appropriate references.

3. Authors should get the services of language editor as some grammatical errors are noticeable in the paper such as:

i. The relationship between a mother and her couple…

ii. …communication and splays mediating roles…

iii. The mediating effect test reveals that peers’ support support…

Reviewer #2: Thank you for the opportunity to review this article. It was very interesting and thought-provoking. The paper explores the relationship between participation in community organizations and mental wellbeing of pregnant mothers in China. In addition, it explores the mediation role of social support (both in-person and online) and communication with one’s partner. Couple’s communication and peer support had direct effects on mental wellbeing. Social media did not. Community participation’s direct effect on mental health depended on the type of social support, however it affected mental wellbeing indirectly through couple’s communication and support as mediators. The authors explained the implications of this research and how it can help mothers in the future.

The authors have clearly highlighted data cannot be openly available except by request due to participant privacy.

Having reviewed the paper, below are some important changes that I think should be considered:

Comment 1: All the mothers in this study are still in gestation which would be considered prenatal, as the title suggests. However, throughout the paper, the word perinatal is used which usually includes mothers for at least one year after birth. For consistency, stick to using “prenatal” except for where past research and literature focused on the wider perinatal period.

Comment 2: The first paragraph lines 38 to 50 do not tell the reader directly what the paper is about. If this study is focusing on non-pathological experiential physical and mental imbalances in prenatal mental health (PMH) (as stated in the last sentence), then mentioning prenatal depression and fear of childbirth could confuse the reader unless these are included in your PMH measure. Furthermore, the paper is focusing on social support and community participation which is the novel aspect of the paper and should be mentioned as early as possible at the end of this paragraph.

Comment 3: The opening paragraph also mentions cultural factors (lines 47 – 49), which although interesting, would be best moved to the beginning of the paragraph at line 62 when highlighting the Chinese context. This would make it clearer that the current paper does not address cultural factors listed but does consider culture in a novel way (for example, ChaxuGeju).

Comment 4: In line 180, it is stated that 1705 responses were “validly answered” but no criteria of what was considered a valid answer is stated. Please state the criteria that deemed a response usable.

Comment 5: For the section on “social support from peers” (line 207), it is unclear what were the social support categories and what items were given for each of the types of support. Line 209 states “such as” which implies that more types of social support were given then the two listed. It is best to state the total number of social supports or if it was just these two, make this clear. In addition, instrumental support is an academic term not always understood by non-experts (at least in English) so how was instrumental support and the other types of support defined to participants?

Comment 6: For your mediation results, confidence intervals would be useful to have for the different effects.

Comment 7: Your tables and path diagrams are really well designed and incredibly useful. I think these are great contributions to the paper.

Comment 8: The discussion is really well written and lines 339 to 399 were particularly well-written and well thought about in relation to the results and their implications, particularly in a Chinese context.

Comment 9: In lines 400, it is claimed that the study “helped mothers in managing various types of support” which is too bold for the paper. The word manage implies some sort of researcher-led intervention or experiment to help mothers which is beyond the aims of this study. The study’s cross-sectional design has more helped our understanding of the importance of community participation and social support for mothers which may have implications for implementing and managing support.

Comment 10: More limitations should be considered. For example, the measure BMSWBI is more about subjective mental wellbeing rather than mental health. Therefore, symptoms of depression or anxiety participants mentioned in the introduction might not have been picked up on. In addition, participants were not asked if their experience of different community organizations was positive or negative. We also do not know what type of support the participants’ felt they received from each organization. Your study also includes only Chinese prenatal mothers and so we do not know if the results are generalisable to mother’s one-year post-birth or mothers in other countries.

In addition to the above comments, I would suggest some minor points for you to consider. Some of these may be considered stylistic choices:

Comment 11: Lines 92, 101, and 216, the word “couple” would be more natural as “partner” in these cases.

Comment 12: Line 150, I think the hypothesis is meant to say “communication play mediating roles”

Comment 13: The descriptive statistics section (Line 242) is not needed as it is covered by Table 1. Lines 243 – 245 are not needed as this is already covered in the participants section. Lines 245 – 247 can be added to the participants section. I would suggest starting the Results with the correlational analyses and mentioning the descriptives can be found in Table 1 within the section.

Comment 14: At the beginning of the correlation section, it would be useful to remind readers what the hypotheses were so readers understand the purpose of these correlations. In addition, you mention when things (partially) support H1, H3, and H4 but do not mention H2 directly in the results.

Overall, this paper would be of interest to the scientific community, and it is clear the authors have worked hard at producing this article. However, the article needs some work on addressing the comments above before it can be published. Therefore, my recommendation for now is minor edits. Thanks again for the opportunity to review and I look forward to seeing a new version in the future.

**Do you want your identity to be public for this peer review?** For information about this choice, including consent withdrawal, please see our Privacy Policy

Reviewer #1: No

Reviewer #2: No

---

## [Author Response · Author response to Decision Letter 1]

2 Jul 2025

1.We appreciate your insightful observation regarding the potential ambiguity associated with the term 'fathers.' We recognize that the original wording lacked precision, which could lead to misunderstandings. In this study, all references to 'fathers' were intended to refer specifically to the partners of the pregnant women—that is, the fathers of the unborn children. To improve clarity and maintain professional accuracy, we have systematically revised the terminology throughout the manuscript. Specifically, the term used in line 98 has been changed to 'partner' to adopt a more inclusive and precise expression, while the term in line 356 has been modified to 'fathers of unborn children' to explicitly specify the intended referent. These revisions aim to ensure consistency and eliminate any ambiguity in the text (see line 115 and line 415).

2.We appreciate your recognition of the value of our research and your observation regarding the inadequacy of references for several claims in the manuscript. We wholeheartedly agree that high-quality academic writing must be firmly rooted in robust and well-documented evidence. In response to your suggestion, we have meticulously reviewed the entire manuscript, paying particular attention to statements that were previously unsupported by citations. We have now enriched these sections with multiple authoritative sources from both domestic and international literature, thereby enhancing the rigor and credibility of our arguments (see lines 40–62, 65–68, 99--100, and 401–406).

3.Thank you for your rigorous attention to linguistic details. We sincerely apologize for the grammatical and lexical issues identified in the manuscript. In response to your comments, we have carefully revised each of the issues you pointed out and conducted a thorough language review of the entire manuscript to improve the clarity and professionalism of our academic writing. The specific revisions are as follows:

(1)The original sentence lacked clarity. It has been revised to: "The relationship between a woman and her partner is frequently regarded as a more significant protective factor in close relationships than those with parents and friends." (refer to lines 109–111)

(2)The original wording was inappropriate. It has been revised to: "Support from peers, whether facilitated through offline groups or social media platforms, serves as a mediating factor in the relationship between community participation and PMH." ( refer to 167–169)

(3)The original sentence contained a redundant repetition ('support support'). It has been corrected to: "The mediating effect test demonstrates that peer support and partner communication serve as mediators in the relationship between community participation and PMH." ( refer to lines 397–399)

4.Thank you for your valuable feedback regarding the inconsistency in terminology usage. We fully concur with your observation that maintaining terminological consistency and scientific accuracy is crucial when describing the study population and research scope. As you rightly pointed out, all mothers who participated in this study were between 13 and 41 weeks of gestation, which falls within the prenatal period. To ensure that our terminology accurately reflects the study population and to prevent any potential confusion for readers, we have systematically reviewed the manuscript. We have replaced all non-referenced uses of the term 'perinatal' with 'prenatal', including in the title and throughout the main text. In instances where the original wording in cited literature uses 'perinatal' while the study content clearly includes the prenatal stage, we have retained the term but added a clarifying note to maintain academic precision.

5.Thank you for highlighting the issue of insufficient clarity in the logical structure of the introduction. In response to your suggestion, we have restructured the opening paragraph to more clearly define the focus of this study. We begin by introducing a dual-dimensional framework of mental health, which distinguishes between pathological imbalance and non-pathological experiential imbalance. We emphasize that the latter is particularly prevalent during the prenatal period, especially in Asian countries such as China, where it is closely tied to sociocultural contexts. While we acknowledge the importance of issues such as prenatal depression and fear of childbirth, we clarify that this study focuses specifically on the non-pathological, subjectively experienced forms of psychological disequilibrium (refer to lines 42–65, and references 5–6).

6.Thank you for your insightful suggestion. As you rightly noted, the key innovation of this study lies in the incorporation of social support and community participation into the analytical framework for understanding prenatal mental health. In the revised manuscript, we have added a statement at the end of the first paragraph to emphasize how social support and community participation serve as moderating and mediating variables that influence the psychological well-being of pregnant women. This addition establishes a conceptual foundation for the subsequent analysis and discussion (refer to lines 53–62,and references 7–11 ).

7.Thank you for your constructive feedback regarding the text structure. Following your suggestion, we have relocated the discussion on cultural factors from the original lines 47–49 to a new position at lines 75–77.

8.Thank you for your insightful comments regarding the imprecise description of the questionnaire data. In response to your suggestion, we have clarified the two criteria for classifying a questionnaire response as 'valid' in lines 203–208: (1) the questionnaire must be fully completed, with no missing items; and (2) the respondent must not display obvious logical inconsistencies or provide deliberate erroneous answers. Furthermore, we have included relevant literature to underscore the significance of questionnaire validity in ensuring the reliability of survey results (refer to lines 203–208,and references 36–37).

9.Thank you for your valuable suggestions regarding the classification and expression of social support. In light of your advice, we have revised the section on "peer social support" as follows. First, this study categorizes social support into three functional types: emotional support, instrumental support, and informational support, encompassing a total of 16 measurement items that cover specific supportive contents within each category. Second, to enhance readability, we have provided concise definitions for each type of support, drawing from the classic framework established by Rodriguez & Cohen (1998). Instrumental support refers to tangible assistance that individuals receive, such as financial aid or help with daily tasks. Informational support involves the provision of relevant knowledge, advice, or guidance aimed at helping individuals cope with challenges or make informed decisions. Emotional support includes expressions of empathy, care, reassurance, and trust, providing individuals with opportunities for emotional expression and psychological relief (see lines 241–253, reference 40).

10.Thank you for pointing out the shortcomings in the presentation of our results. In response, we have added the 95% confidence intervals for each major effect path in Table 3 to enhance the transparency and interpretability of our findings. (See lines 378–379)

11.Thank you for highlighting the potential for misunderstanding in our original wording. We fully agree that the cross-sectional design of this study does not allow for direct inference of intervention effects. Accordingly, we have revised the relevant statement to more accurately reflect the study’s aims and findings. The text now reads: “This study aims to examine the role of community involvement and social support in the lives of mothers, providing insights that may inform the design and implementation of supportive interventions.” (See lines 458–460)

12.Thank you very much for your thoughtful and constructive feedback. We sincerely appreciate your careful review and fully acknowledge the valuable concerns you have raised. In response, we have revised the Limitations section of the manuscript accordingly (see lines 460–475) and would like to provide the following detailed responses. We acknowledge that the Body–Mind–Spirit Well-Being Inventory (BMSWBI) primarily captures subjective and culturally grounded dimensions of psychological well-being and may not comprehensively detect clinical manifestations of mental disorders such as depression or anxiety. As our study focused on experiential imbalance—a culturally nuanced conceptualization of mental distress rather than clinical diagnosis—the BMSWBI was selected for its alignment with our theoretical framework and its validated application in Asian populations. Nonetheless, we agree that the absence of standardized diagnostic tools such as the PHQ-9 or GAD-7 limits our ability to identify clinical symptomatology. This limitation is now explicitly acknowledged in the revised manuscript; alongside a recommendation that future research incorporate such instruments to further explore the intersections between culturally defined well-being and clinically defined mental health conditions. (See lines 460–475)

13.Thank you very much for your thoughtful and constructive feedback. We sincerely appreciate your careful review and fully acknowledge the valuable concerns you have raised. In response, we have revised the Limitations section of the manuscript accordingly (see lines 460–475) and would like to provide the following detailed responses. We fully agree that subjective evaluations of experiences with community organizations—whether positive, neutral, or negative—are essential for understanding the impact of community participation on well-being. While our study measured participants’ involvement across various community groups, it did not capture their perceptions of the quality of those interactions, nor did it identify specific organizations or the distinct types of support they provided. This significant limitation has been explicitly acknowledged in the revised manuscript. Although we were unable to examine all types of organizations in detail, our data did assess participants’ perceived emotional, instrumental, and informational support from key domains such as family, healthcare, peer groups, community, government, and social media. These indicators offer a multidimensional understanding of social support that complements the broader analysis presented in our findings. The implications of these findings will also be reflected in other research outcomes. (See lines 460–475)

14.Thank you very much for your thoughtful and constructive comments. We sincerely appreciate your careful review and fully acknowledge the valuable concerns you have raised. In response, we have revised the Limitations section of the manuscript accordingly (see lines 460–475) and would like to provide the following detailed responses. We appreciate your observation regarding the demographic scope of our sample. Since our study focuses specifically on prenatal mothers in China, the findings may not be generalizable to postpartum mothers or to mothers in different cultural contexts. We have clarified this limitation in the manuscript. Additionally, we would like to highlight that a parallel longitudinal qualitative study is currently underway, tracking a subset of 83 participants from pregnancy through six months postpartum. This ongoing research will enable us to investigate the evolution of maternal well-being and support dynamics throughout the perinatal period. Future also includes cross-cultural comparisons to assess the consistency of these patterns across diverse sociocultural settings. (See lines 460–475)

15.Thank you for your suggestion regarding the use of terminology. In response, we have revised the manuscript to replace “couple” with the more appropriate and inclusive term “partner” in all relevant instances.

16.Thank you for your insightful observation. Based on your feedback, we have revised the hypothesis statement to more clearly reflect the mediating role of peer support. The updated sentence now reads: “Support from peers, whether facilitated through offline groups or social media platforms, serves as a mediating factor in the relationship between community participation and PMH.” (see lines 167–169)

17.We appreciate your insightful and constructive suggestion. We fully concur with your observation that repeating descriptive statistics already presented in Table 1 within the Results section leads to unnecessary redundancy. In response to your recommendation, we have made the following revisions: First, we have relocated the descriptive statistics originally presented in the Results section (line 242) to the 'Participants' subsection, thereby consolidating the presentation of sample characteristics. This restructuring enhances the clarity and logical coherence of the manuscript (see lines 211–216).Second, following your advice, we have introduced a correlation analysis at the beginning of the 'Results' section to underscore the statistical relationships among the dependent, independent, and mediating variables. This addition improves the logical flow leading into the subsequent model testing (see lines 309-323). Finally, we have explicitly indicated in the text that Table 1 contains the complete set of descriptive statistics to prevent redundancy.

18.Thank you very much for your valuable suggestion. We fully agree that restating the core hypotheses and key variables at the beginning of the correlation analysis section enhances readers’ understanding of the purpose and logical structure of the analysis. In response to your recommendation, we have revised the opening paragraph of the correlation analysis section to explicitly restate the main hypotheses of the study and clearly define the dependent, independent, and mediating variables. This revision improves the readability and coherence of the section, thereby establishing a stronger theoretical foundation for the subsequent structural equation modeling and path analysis (see lines 285–294).

19.Thank you for bringing to our attention the omission of a response regarding H2 in our article; this was indeed an oversight on our part. We have now included a comprehensive response to H2 in the results section, specifically on line 349.

20.Thank you for your thorough review and valuable guidance regarding the manuscript's adherence to PLOS ONE's formatting and ethical standards. We have meticulously addressed each of your points as follows: (1)We have revised the manuscript and all accompanying files to ensure compliance with PLOS ONE's formatting and file naming requirements. (2) The corresponding author has successfully registered and validated their ORCID id in the Editorial Manager system, following the provided instructions.(3) We have added a comprehensive ethics statement in the Methods section of the revised manuscript. This now includes the full name of the ethics committee that approved our study, the approval reference number, and a clear explanation of the informed consent process. (see revised Methods section, lines 196–201). (4) In accordance with PLOS ONE's data sharing policy, we have made all data underlying the findings of this study available within the manuscript itself. (see lines 507-509) (5) Personal identifiers have been removed to ensure participant confidentiality, and all shared data comply with the terms of informed consent obtained from participants. (6) All Supporting Information files have now been properly captioned and listed at the end of the manuscript. In-text citations referencing these files have also been updated for consistency and clarity (see lines 503–509). (7) We have thoroughly reviewed the reference list to ensure that all citations are complete, accurate, and formatted according to PLOS ONE style. Missing or incomplete entries have been corrected (see lines 511–681).

---

## [Decision Letter · Decision Letter 1]

16 Jul 2025

Dear Dr. Sun,

Thank you for submitting your manuscript to PLOS ONE. After careful consideration, we feel that it has merit but does not fully meet PLOS ONE’s publication criteria as it currently stands. Therefore, we invite you to submit a revised version of the manuscript that addresses the points raised during the review process.

We look forward to receiving your revised manuscript.

Kind regards,

Fucai Lin, Ph.D.

Academic Editor

PLOS ONE

Journal Requirements:

Reviewers' comments:

Reviewer's Responses to Questions

**Comments to the Author**

Reviewer #1: (No Response)

Reviewer #2: All comments have been addressed

2. Is the manuscript technically sound, and do the data support the conclusions?

Reviewer #1: Yes

Reviewer #2: Yes

3. Has the statistical analysis been performed appropriately and rigorously?

Reviewer #1: Yes

Reviewer #2: Yes

4. Have the authors made all data underlying the findings in their manuscript fully available?

Reviewer #1: Yes

Reviewer #2: No

5. Is the manuscript presented in an intelligible fashion and written in standard English?

Reviewer #1: Yes

Reviewer #2: Yes

Reviewer #1: The study investigated an important but often neglected determinant of maternal and child health outcomes. The methodology is sound and detailed, however, the quality can be improved further. My observations are below.

1. The abstract states that 'a cross-sectional survey was conducted…' but no mention of this in the full text.

2. As you indicated, 'Maternal Depression Generation Trajectory and Social Support Research,' included demographic data'. Why was this omitted in the findings? Kindly indicate results about the participants' demography both in the abstract and main text?

3. Please confirm that all typographical errors have been corrected. Check lines 121-122: "the rise of the social media and social media…"

4. Indicate if the survey questionnaire, the 'Maternal Depression Generation Trajectory and Social Support Research,' had been pretested prior to the study. Or was it originally designed for this study? You mentioned the source of some survey instruments used, please do the same for all instruments.

5. Indicate how long it took to complete the survey.

6. Please report the challenges that were encountered in the course of the study.

7. Did you consider if parity could have mediated any of the findings? Did support differ by parity? e.g. did first time mothers have more support than second or third timers?

8. Since participants were recruited from 3 different facilities, did your findings differ based on location or was the analysis aggregated? It would be interesting to see the differences and similarities by location.

9. Are the study sites urban or rural? Don’t you think this could have influenced the availability, type and quality of support available to pregnant women?

10. What type of incentives/small gift was provided for the participants? When was this provided? Before or after filling out the survey? You should detail this to show that undue inducement was avoided.

11. You indicated that "Eligible participants were guided by the research team to view the online research information form and consent form…" You also need to explain the mode of instrument administration. Was it by the researcher, participants or by who? This is important for replicability.

13. Describe the rationale for the sample size selection. What method of sampling was utilized? Was it based on population size or what? How did you arrive at the total (1899)?

14. You should add a separate section for limitations of your study rather than its current location within the discussion. Add other limitations to what is already reported. E.g., the study could have benefited from triangulation of methods and participants. 1) only pregnant women were sampled without considering the perspectives of their social networks, such as partners. 2) Also only survey was used in the study, using a mixed methodology could have enriched the findings further, etc.

Reviewer #2: Thank you for submitting your paper once again. The paper has improved greatly and it is much easier to follow the line of thought in the introduction. The methods, results, and discussion are now very clear and I hope this paper encourages future work on prenatal mental wellbeing. I applaud the authors for their hard work and am recommending an accept for the manuscript.

Although I think the paper is in good shape, clarification is needed around data availability. By data availability, we mean access to the datasheet (e.g., Excel, CSV file, transcripts) and not to the statistics from analyses. Authors currently do not provide the data itself so the availability of the data statement is incorrect. Authors would need to upload the anonymized datafile as a supplementary material in order for the current statement to be correct.

My understanding from last review is that there were restrictions for the data. If this is still the reason for being unable to share data then I would recommend for authors to use the statement below instead (adding an email address to the ethics committee and not the author), or an alternative wording recommended by the editor:

"Data cannot be shared publicly because of data contain potentially identifying and sensitive patient information. Data are available from the Medical Ethics Committee of Jiangnan University

(JNU20211217IRB01) (contact via INSERT EMAIL) for researchers who meet the criteria for access to confidential data."

**Do you want your identity to be public for this peer review?** For information about this choice, including consent withdrawal, please see our Privacy Policy

Reviewer #1: No

Reviewer #2: **Yes: ** Philip Howlett

---

## [Author Response · Author response to Decision Letter 2]

5 Aug 2025

Reviewer 1

1.The abstract states that 'a cross-sectional survey was conducted…' but no mention of this in the full text.

Thank you for highlighting the deficiencies in our report results. In response, we have incorporated the language related to the cross-sectional survey into the methodology and limitations of the research design.

In the methods section, we added the following statement: "We conducted a cross-sectional survey on a representative sample of pregnant women in China". (see lines 181-182)

Additionally, in the limitations section, we included "the COVID-19 quarantine policy has significantly hindered our ability to collect cross-regional and large-scale data. Consequently, this study utilized regional samples and gathered data in clinical settings, which may limit our capacity to observe the evolution of mental health and support throughout pregnancy and the postpartum period ".

(see lines 508-512)

2.Kindly indicate results about the participants' demography both in the abstract and main text?

Thank you for your insightful feedback. In the revised main manuscript section, we have taken into account the comments from the first round of reviewers and adjusted the description of demographic characteristics from the results section to the participants section. However, as you rightly pointed out, we need to further enhance the presentation of this content to provide other researchers with a clearer understanding of the study's research subjects. In response to your suggestions, we have included additional details regarding demographic characteristics in the summary section, such as the average age, parity, registered residence of participants, the average number of community participations, and the average mental health score. Furthermore, we have added a table of demographic characteristics based on the existing data in the participants section. (see lines 231-237 , and Table 1)

3.Please confirm that all typographical errors have been corrected. Check lines 121-122: "the rise of the social media and social media…"

Thank you for your rigorous attention to linguistic details. In response to your comments, we carefully revised each of the issues you pointed out and conducted a thorough language review of the entire manuscript to improve the clarity and professionalism of our academic writing.

4.Indicate if the survey questionnaire, the 'Maternal Depression Generation Trajectory and Social Support Research,' had been pretested prior to the study. Or was it originally designed for this study? You mentioned the source of some survey instruments used, please do the same for all instruments.

Thank you for your insightful comment. Based on your feedback, we would like to provide further clarification regarding the “Maternal Depression Generation Trajectory and Social Support Research” questionnaire. We appreciate the opportunity to elaborate on this point and hope the following explanation addresses your concerns.

Questionnaire Design. The questionnaire was originally designed for this study, based on a specific theoretical framework and research questions concerning the generative mechanisms of maternal depression and its relationship to social support.

Pretesting. The present study employs a survey questionnaire as a method for data collection. Prior to executing the pretest, the questionnaire was finalized, and a pretest form was prepared using the framework from Hu, S. (2023)[1]. The survey questionnaire was administered face-to-face in the hospital to facilitate the distribution of both the questionnaire and the pretest form. The stages undertaken during the pretest are as follows:

1. Selecting the Instrument: The researcher clearly defined each construct within the framework in a manner consistent with past research. This approach was informed by a literature review of previous theoretical and empirical studies on the constructs, as recommended.

2. Expert Review for Validity: This stage was conducted prior to the distribution of the survey questionnaire to ensure the validity of the instruments. The researcher selected and consulted three experts in the field of maternal public mental health; among them, there was one midwife, one mental health professional, and one social medical expert.

3. Application for Ethics Committee Approval: The researcher applied for ethics committee approval before the data collection phase.

4. Selecting Respondents: At this stage, the researchers recruited 30 pregnant women to participate in the questionnaire survey at the hospital with the highest number of births per year.

5. Executing the pretest. The questionnaire was distributed face-to-face in this prediction test by researchers to participants, and was subsequently collected after all participants had completed their responses. The prediction test form consists of three parts: the first part is the cover letter, the second part is the questionnaire, and the third part is the prediction test form. The cover letter briefly introduces the research background, the purpose of the prediction test, the implementation process, and guarantees the anonymity and confidentiality of the respondents. The second part of the questionnaire contains specific topics related to each variable, while the prediction test requests respondents to provide feedback on the questions and scales in the form of queries and to suggest improvements.

6. Review results. SPSS software was employed to assess the validity and reliability of the collected data. It was determined that the items in the instrument met the minimum acceptable requirement of a Cronbach's alpha of 0.7. Feedback from participants regarding unclear terminology and ambiguous expressions in individual items was addressed and corrected.

Source of Instruments. As you rightly pointed out, some scales or items were adapted from existing validated instruments. Below is a complete list of all instruments used, along with their sources: (1) Mother's PMH. This questionnaire was utilized to assess the mental health of Chinese mothers, specifically the revised Body Mind-Spirit Well-Being Inventory (BMSWBI) [2]. (2) Community Participation. This variable follows Lin's research methodology to assess community connections through involvement in community organizations [3]. (3) Social Support from Peers. This questionnaire measured the support received from offline peer groups, specifically pregnant women and social media, utilizing the framework from Olson et al. [4]. (4) Partners’ Communication. The Initiator Style Questionnaire (ISQ), developed by Denton and Burleson [5], prompts mothers to assess their inclination towards initiating or avoiding discussions about relationship issues with their partners.

In response to your comment, we have revised the manuscript to clearly indicate the origin of each instrument and noted the pretest process in the methods section. We have added this information to the methods section of the manuscript. (see lines 189-196, and references 16,38,40-41)

5.Indicate how long it took to complete the survey.

Thank you for your valuable comment. We have included the complete duration for the survey in the methods section of the manuscript. Specifically, we state: "On average, participants took approximately 20-30 minutes to complete the survey. This duration was recorded during the pretest phase and reconfirmed during the formal data collection process". (see lines 204-205)

6.Please report the challenges that were encountered in the course of the study.

Thank you for your valuable comment. As you have rightly pointed out, we encountered several challenges during the investigation process. However, due to the manuscript's word limit, we were unable to elaborate on these details in the main text. We sincerely appreciate your understanding and would like to provide a clearer explanation in this document.

Challenges 1 Participant Recruitment. Given the sensitive nature of maternal mental health, some potential participants were initially hesitant to engage in the survey. To address this issue, we adopted a two-pronged approach. First, we displayed posters in the obstetrics clinics of each participating hospital to introduce the purpose and significance of the study, thereby promoting transparency and ensuring informed consent. Second, we distributed the questionnaires during the waiting periods for routine prenatal check-ups, a time when participants were relatively at ease and more likely to have the time and willingness to thoughtfully complete the survey. This approach helped reduce refusal rates and improved the quality and completeness of the data collected.

Challenges 2 Data Collection in Clinical Settings. Conducting research within hospital settings posed a significant challenge due to the requirement for multi-level administrative approvals to access specific departments. To overcome this barrier, we established formal collaborations with the research administration offices of each participating hospital. These institutional partnerships not only expedited the approval process but also enhanced the study's credibility among clinical staff. Consequently, we were able to build trust with healthcare professionals, which facilitated smoother and more efficient data collection.

Challenges 3 COVID-19 Related Disruptions. Intermittent restrictions during the pandemic delayed in-person visits and necessitated adjustments in recruitment and scheduling. The investigation coincided with the COVID-19 epidemic, which required adherence to China's epidemic prevention policies. Upon identifying a suspected case, quarantine measures were implemented. During the investigation, a suspected case emerged in the obstetrics inpatient departments of Hospital B, where we were conducting our research. Consequently, all obstetric medical staff, pregnant women, and some researchers were transferred, forcing the research team to halt the obstetrics investigation for two months. This experience of navigating the COVID-19 epidemic alongside the obstetric medical team and pregnant women has, to some extent, strengthened our trust and facilitated further research and investigation.

Thank you for your guidance, which will assist us in further optimizing and enhancing the content of the limitations sections. Due to the word limit of the main manuscript, we have included some of the content from the challenges in the limitations section. (see lines 494-514)

7.Did you consider if parity could have mediated any of the findings? Did support differ by parity? e.g. did first time mothers have more support than second or third timers? Thank you for raising an important point regarding parity as a potential factor of study findings, particularly concerning the differences in support received by first-time versus experienced mothers. To address this, we conducted a one-way ANOVA comparing indicators of psychological well-being and social support across three parity groups, first-time mothers, second-time mothers, and those with three or more pregnancies.

Our analysis revealed that most psychological indicators—such as maternal positive mental health (PMH), body, mind, spirit, and informational support—did not differ significantly by parity (all p > .05). However, several key support-related outcomes exhibited significant differences. Specifically, first-time mothers reported significantly higher levels of community participation (F = 13.56, p < .001), social support (F = 13.46, p < .001), emotional support (F = 12.92, p < .001), and instrumental support (F = 26.81, p < .001) compared to those with previous pregnancies. Post-hoc analyses (with LSD) indicated small to moderate effect sizes between first-time and multiparous mothers; for instance, instrumental support was notably higher among first-time mothers compared to both second-time (d = 0.337*) and third-or-more-time mothers (d = 0.538*). These findings suggest that first-time mothers may benefit from heightened levels of social attention and support. In contrast, mothers with multiple pregnancies may receive relatively less support, which could have implications for designing targeted interventions.

We sincerely appreciate your suggestion once again, as this analysis offers valuable insights into how parity status may influence maternal support experiences. Given the word limit constraints of the main manuscript, we have summarized these parity-based findings in a supplementary table and briefly noted them in the "Correlation analysis between the main variables" section.

(see lines 337-341 in the manuscript, and Appendix 2 at the end of this document)

8.Since participants were recruited from 3 different facilities, did your findings differ based on location or was the analysis aggregated? It would be interesting to see the differences and similarities by location.

Thank you for raising an important point regarding facilities as a potential factor influencing study findings, particularly concerning the differences in support received by the three hospitals, DH-A, DH-B, and DH-C. To address this, we conducted a one-way ANOVA to compare key PMH and support-related variables across these three facilities, followed by post-hoc comparisons to further investigate specific group differences.

Our analysis revealed that mother's PMH, mind, spirit, social support, emotional support, and informational support did not show statistically significant differences across hospitals (all p > .05). This suggests a relative consistency in these domains across locations. In contrast, body-related well-being (F = 3.89, p = .02) and instrumental support (F = 3.10, p = .05) exhibited modest but significant differences by location. Post-hoc analyses (with LSD) indicated that DH-C participants reported higher levels in both indicators compared to DH-A and DH-B (e.g., body: DH-C > DH-B, d = 0.145*, DH-C > DH-A, d = 0.176*; instrumental support: DH-C > DH-A, d = -0.169*). Notably, community participation demonstrated the most marked difference across hospitals (F = 9.70, p < .001), with participants at DH-C reporting significantly greater involvement than those at DH-A (d = -0.662*) and DH-B (d = -0.286). These differences may reflect variations in outreach programs at C hospital or regional community engagement levels.Overall, while the majority of psychosocial indicators were relatively consistent across sites, a few context-sensitive variables, such as community participation and instrumental support, exhibited statistically significant location-based variation. These findings suggest that while aggregated analysis was appropriate for many variables, site-specific differences should be noted and may warrant further exploration in future research.

We sincerely appreciate your suggestions once again, as this analysis provides valuable insights into how the status of different facilities may influence maternal support experiences. Given the word limit constraints of the main manuscript, we have summarized three facility-based findings in a supplementary table and briefly noted them in the "Correlation Analysis Between the Main Variables" section.

(see lines 333-337 in the manuscript and Appendix 3 at the end of this document).

9.Are the study sites urban or rural? Don’t you think this could have influenced the availability, type and quality of support available to pregnant women?

Thank you for highlighting the critical issue of the urban-rural background of the research site and its potential impact on the availability, type, and quality of support for pregnant women.The three hospitals involved in the study are situated in urban areas of Jiangsu Province. However, Jiangsu is a highly urbanized region that has seen a substantial influx of rural populations migrating to cities, particularly in the surveyed city of Wuxi (WX), which experiences a significant floating population. According to statistical data, the permanent population of the surveyed cities is 7.505 million, with an urbanization rate of 86.61%. This includes a registered population of 5.2294 million and a non-registered population of 2.2756 million.Consequently, the obstetrics wards of these hospitals cater to diverse populations, including urban residents, rural residents, and rural women from other pro

---

## [Decision Letter · Decision Letter 2]

4 Sep 2025

Dear Dr. Sun,

Thank you for submitting your manuscript to PLOS ONE. After careful consideration, we feel that it has merit but does not fully meet PLOS ONE’s publication criteria as it currently stands. Therefore, we invite you to submit a revised version of the manuscript that addresses the points raised during the review process.

https://journals.plos.org/plosone/s/submission-guidelines#loc-laboratory-protocols . Additionally, PLOS ONE offers an option for publishing peer-reviewed Lab Protocol articles, which describe protocols hosted on protocols.io. Read more information on sharing protocols at https://plos.org/protocols?utm_medium=editorial-email&utm_source=authorletters&utm_campaign=protocols .

We look forward to receiving your revised manuscript.

Kind regards,

Fucai Lin, Ph.D.

Academic Editor

PLOS ONE

**Journal Requirements:**

**Additional Editor Comments:**

The study investigated an important but often neglected determinant of maternal and child health outcomes. The methodology is sound and detailed, however, the quality can be improved further. My observations are below.

1. The abstract states that 'a cross-sectional survey was conducted…' but no mention of this in the full text.

2. As you indicated, 'Maternal Depression Generation Trajectory and Social Support Research,' included demographic data'. Why was this omitted in the findings? Kindly indicate results about the participants' demography both in the abstract and main text?

3. Please confirm that all typographical errors have been corrected. Check lines 121-122: 'the rise of the social media and social media…'

4. Indicate if the survey questionnaire, the 'Maternal Depression Generation Trajectory and Social Support Research,' had been pretested prior to the study. Or was it originally designed for this study? You mentioned the source of some survey instruments used, please do the same for all instruments.

5. Indicate how long it took to complete the survey.

6. Please report the challenges that were encountered in the course of the study.

7. Did you consider if parity could have mediated any of the findings? Did support differ by parity? e.g. did first time mothers have more support than second or third timers?

8. Since participants were recruited from 3 different facilities, did your findings differ based on location or was the analysis aggregated? It would be interesting to see the differences and similarities by location.

9. Are the study sites urban or rural? Don’t you think this could have influenced the availability, type and quality of support available to pregnant women?

10. What type of incentives/small gift was provided for the participants? When was this provided? Before or after filling out the survey? You should detail this to show that undue inducement was avoided.

11. You indicated that 'Eligible participants were guided by the research team to view the online research information form and consent form…' You also need to explain the mode of instrument administration. Was it by the researcher, participants or by who? This is important for replicability.

13. Describe the rationale for the sample size selection. What method of sampling was utilized? Was it based on population size or what? How did you arrive at the total (1899)?

14. You should add a separate section for limitations of your study rather than its current location within the discussion. Add other limitations to what is already reported. E.g., the study could have benefited from triangulation of methods and participants. 1) only pregnant women were sampled without considering the perspectives of their social networks, such as partners. 2) Also only survey was used in the study, using a mixed methodology could have enriched the findings further, etc.

Reviewers' comments:

Reviewer's Responses to Questions

**Comments to the Author**

Reviewer #1: All comments have been addressed

Reviewer #2: All comments have been addressed

2. Is the manuscript technically sound, and do the data support the conclusions?

Reviewer #1: Yes

Reviewer #2: Yes

3. Has the statistical analysis been performed appropriately and rigorously?

Reviewer #1: Yes

Reviewer #2: Yes

4. Have the authors made all data underlying the findings in their manuscript fully available?

Reviewer #1: No

Reviewer #2: Yes

5. Is the manuscript presented in an intelligible fashion and written in standard English?

Reviewer #1: Yes

Reviewer #2: Yes

**Reviewer #1:**  All comments have been addressed, though some were not included in the manuscript due to word count limitation.

**Reviewer #2:**  The authors have worked hard on the paper and have addressed my comments in both rounds of revisions. The data availability statement is much clearer.

A table of demographics has also been provided and also reasoning of sample size (using G*Power) which make for a more transparent methodology. The authors have also done extra analyses regarding the parity of mothers and across locations in response to reviewer one. The differences and similarities in first-time mothers versus other mothers is interesting for future research and is a nice addition to the paper.

Overall, I think the paper is in excellent shape.

**Do you want your identity to be public for this peer review?** For information about this choice, including consent withdrawal, please see our Privacy Policy

Reviewer #1: No

Reviewer #2: **Yes: ** Philip Howlett

---

## [Author Response · Author response to Decision Letter 3]

10 Sep 2025

Responses to Academic Editor’s comments on Journal Requirements

1.If the reviewer comments include a recommendation to cite specific previously published works, please review and evaluate these publications to determine whether they are relevant and should be cited. There is no requirement to cite these works unless the editor has indicated otherwise.

Response: We sincerely appreciate the clarification provided by the Academic Editor regarding the citation of previously published works. After carefully reviewing the feedback from all reviewers, we found no explicit requirements to cite additional publications. Nonetheless, in the first round of revisions, we incorporated several new references in direct response to the reviewers’ comments. These additions were explained in our initial rebuttal letter and subsequently acknowledged by the reviewers during the second round of review. Should any further citation suggestions arise in future communications, we will carefully assess their relevance and incorporate them where appropriate.

Response: We would like to express our sincere gratitude to the Academic Editor for the insightful feedback on the references, which has significantly strengthened the alignment of our manuscript with academic standards and rigor. In response, we meticulously reviewed the entire reference list to ensure its completeness and accuracy. Furthermore, both authors independently conducted thorough checks using the Web of Science and PubMed databases, in addition to consulting reliable resources such as Retraction Watch and publishers’ official notices. These checks confirmed that none of the cited works have been retracted or flagged for retraction.We hope this reassures you of the reliability and academic integrity of our references, and we remain fully committed to maintaining the highest scholarly standards in our work. Should any updates arise in the future, we will promptly make the necessary revisions.

---

## [Editor Report · Decision Letter 3]

14 Sep 2025

Community participation enhanced prenatal mental health through strengthening peers’ support and partners’ communication in Chinese mothers: A cross-sectional study

PONE-D-24-56566R3

Dear Dr. Sun,

We’re pleased to inform you that your manuscript has been judged scientifically suitable for publication and will be formally accepted for publication once it meets all outstanding technical requirements.

Kind regards,

Fucai Lin, Ph.D.

Academic Editor

PLOS ONE
---

## [Editor Report · Acceptance letter]

PONE-D-24-56566R3

PLOS ONE

Dear Dr. Sun,

I'm pleased to inform you that your manuscript has been deemed suitable for publication in PLOS ONE. Congratulations! Your manuscript is now being handed over to our production team.

Kind regards,

on behalf of

Professor Fucai Lin

Academic Editor

PLOS ONE